# Femtosecond Laser Direct Writing of Gradient Index Fresnel Lens in GeS₂-Based Chalcogenide Glass for Imaging Applications

**Pierre Delullier** [1,2,*][ID]**, Guillaume Druart** [1]**, Florence De La Barrière** [1]**, Laurent Calvez** [3] **and Matthieu Lancry** [2]

[1] ONERA, The French Aerospace Lab, 91120 Palaiseau, France; guillaume.druart@onera.fr (G.D.); florence.de_la_barriere@onera.fr (F.D.L.B.)
[2] ICMMO, Institut de Chimie Moléculaire et des Matériaux d'Orsay, Université Paris-Saclay, 91405 Orsay, France; matthieu.lancry@universite-paris-saclay.fr
[3] Institut des Sciences Chimiques de Rennes-UMR6226, Université Rennes, 35000 Rennes, France; laurent.calvez@univ-rennes1.fr
[*] Correspondence: pierre.delullier@onera.fr

**Abstract:** Chalcogenide glasses have attracted growing interest for their potential to meet the demands of photonic applications in the Mid-Wavelength InfraRed (MWIR) and Long-Wavelength InfraRed (LWIR) transmission windows. In this work, we investigated the photosensitivity to femtosecond laser irradiation of a dedicated chalcogenide glass, along with its possible applications in micro-optics. In order to address the SWaP problem (Size, Weight and Power), this work took advantage of recent techniques in femtosecond laser direct writing to imprint flat and integrated optical systems. Here, we wanted to simplify an infrared multispectral imaging system which combines a lens array and a filter array. Each channel has a focal length of 7 mm and an f-number of 4. We show in this paper that the chosen GeS₂-based chalcogenide glass is very promising for the fabrication of graded index optics by fs-laser writing, and particularly for the fabrication of Fresnel lenses. We note a very important phase variation capacity in this infrared material corresponding to refractive index variations up to +0.055. A prototype of Fresnel GRIN lens with a refractive index gradient was fabricated and optically characterized in the Vis range.

**Keywords:** optical glasses; chalcogenide; optical components; femtosecond laser writing; IR optics

## 1. Introduction

The SWaP problem (Size, Weight and Power) is an important challenge in infrared imaging applications today. In embedded optics, reducing the size, mass and power of systems while maintaining their quality is essential for their integration into ever-smaller carriers (optronic balls, micro-drones, etc.). In order to simplify the optical system before the sensor and to maintain a high image quality, optical designers have to find new degrees of freedom in their designs. Several avenues have been thus explored, such as freeform optics [1], GRIN (gradient index) optics [2] and metasurfaces [3]. GRIN optics involve creating a spatial variation of the refractive index in the material. In its primary use, this index variation can correct some optical aberrations such as spherical aberration [4] and chromatic aberrations [5–7], or can encode optical power inside the lens (not simply at the surface) [8,9].

One of the main limitations of GRIN optics is their manufacturing [10]. Most fabrication techniques can create typical index profiles such as axial or radial [11]. However, freeform or even aspheric index profiles are much more challenging to implement [10]. Alternative solutions exist, such as using materials transparent to the visible range and coated with thin films [12]. However, this method is restricted in terms of material choice [10]. In this work, we focused on femtosecond laser direct writing (FLDW), which is, to date,

the most flexible technique with respect to both inscribed shape and refractive index modulation amplitude and profile.

More specifically, FLDW is a technique for locally modifying the physicochemical properties of amorphous and crystalline transparent materials. It consists of focusing a femtosecond laser beam at, or under, the surface of a material, typically from a few μm up to a few cm [13]. The focused laser light can thus reach intensities higher than tens of TW/cm$^2$ inside the material. At these intensities, nonlinear light absorption phenomena become preponderant, including multiphoton photoionization and tunneling effects. This ultimately yields to local and permanent modifications of the glass, strongly confined in the focal volume on a few μm$^3$ around the laser focal point. Because of its flexibility in terms of both accessible materials and writing possibilities [14], it is a very attractive technique for the fabrication of complex 3D photonic components in a single step. It has been particularly studied for imprinting gratings and optical waveguides [15]. However, FLDW also has attracted the attention of researchers in other photonics applications such as diffractive [16] and polarization [17] optics, gradient index optics [18] and 3D geometric phase optics [19].

FLDW has been extensively studied over the last two decades in silica glass [20], which offers excellent optical properties in the visible and near-infrared range, and more recently in silicon [17]. However, investigations in other materials remain quite limited, especially those to be used in the Mid-Wavelength InfraRed (MWIR 3–5 μm) and Long-Wavelength InfraRed (LWIR 8–14 μm) ranges. It would be of practical importance for applications including, but not limited to, home automation, smartphones and even automotive applications. The few studies of infrared glasses mainly dealt with the fabrication of waveguides [13,21,22] and did not address the fabrication of free space optics or micro-optics for imaging applications. In order to reach these longer wavelengths in the IR range, nonsilicate glasses such as chalcogenide [23], fluoride [24] or heavy oxide [25] glasses must be used. All these glass families have a wider transmission band than the ubiquitous silicate glasses up to the mid-IR range. More specifically, chalcogenide materials have the advantage of exhibiting a very wide transmission band which can, for some specific compositions, span the visible and LWIR windows. These glass materials are increasingly being used in the field of infrared optical imaging [6] for their excellent optical qualities but remain infrequently studied in integrated optical component manufacturing, particularly with the FLDW technique.

This paper is devoted to the design and fabrication by fs-laser writing of a GRIN lens in a GeS$_2$-based chalcogenide glass in order to simplify a multispectral camera by merging the lens array with a filter array. This is a follow-up to early work [26] wherein we made a preliminary design of a GRIN Fresnel lens to simplify a multispectral camera. In the present work, we added detailed optical modeling and a complete phase variation characterization, together with GRIN Fresnel lens fabrication and its characterization. First, we present the design of the gradient index Fresnel lens used to simplify the multispectral camera. We then studied the photosensitivity to IR fs-laser irradiation of $75(GeS_2) - 15(In_2S_3) - 10(CsCl)$ chalcogenide glass that is suitable for targeted applications. Finally, we fabricated and characterized a first gradient index Fresnel lens prototype.

## 2. A gradient Index Plate in a Multispectral Camera

### 2.1. Multispectral Camera

In recent years, the ONERA laboratory has been developing ultracompact and very high-performance infrared cameras. To provide a very low noise temperature difference (so-called NETD) for the whole camera, including the IR detector and the optics, we have developed cryogenic multichannel cameras [27–30]. The latest example is a multispectral camera, for which the concept is illustrated in Figure 1. It consists of an array of 2 × 2 lenses with a focal length of 7 mm and an F-number of 4. Each channel has a field of view of 40° × 30° and forms a 320 × 256 pixel image on each quarter of the detector. An array of 2 × 2 filters is placed after the lens array, so that each optical channel corresponds to a different spectral band with a width of about 200 nm. Cooling the lens array and the

filters allows a very low instrumental flux to be obtained. This multispectral camera has a NETD close to that of the IR detector itself to ensure high performance levels. This kind of camera can be used, for example, in the field of gas leak detection and quantification in industrial installations.

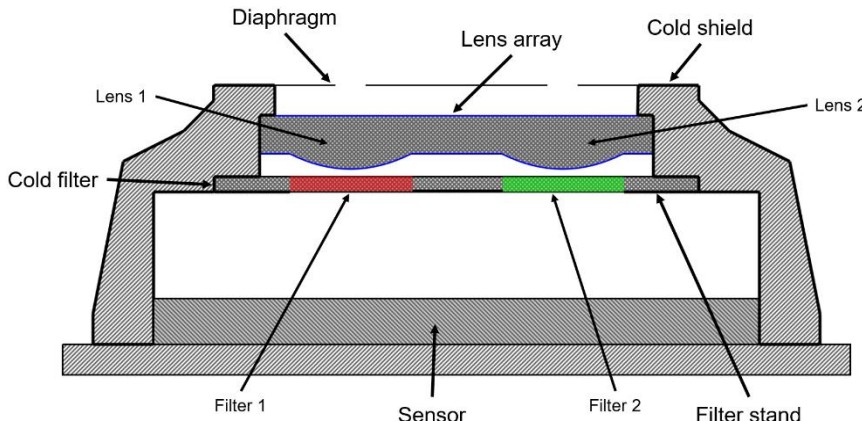

**Figure 1.** Cross-section of the ONERA multispectral camera. The diaphragm, the lens array, the filter array and the sensor are shown from top to bottom.

The optical system is thus composed of only two optical objects, a lens array for imaging and a filter array to select a specific spectral band for each channel. These components are integrated into the cold shield of a Detector Dewar Cooler Assembly (DDCA) and are cooled to cryogenic temperature. The drawback of adding optical components inside the DDCA is that it increases the thermal mass of the cold shield and this tends to increase the cool-down time of the DDCA. In order to limit the additional mass to be cooled, we wanted to remove one of the components. The filter array is made of interference filters composed of multiple thin layers deposited on a substrate. It is challenging to deposit these films on a nonplanar surface such as the lens array. Consequently, a viable solution would be to use a GRIN optic to integrate both focusing and filtering functions in the same flat and thin glass plate, achieving the expected mass reduction. In this work, we studied only one channel of this multichannel camera.

A GRIN focusing plate needs radial index variation to create optical power instead of a spherical surface. A classical index profile used in this case is described in Equation (1):

$$n(r) = n_0 + n_r r^2 \qquad (1)$$

where $n_0$ is the refractive index on the optical axis, r the radial position and $n_r$ the coefficient of index variation. This refractive index distribution gives to the optical plate an optical power V [31]:

$$V = \frac{1}{f} = -2n_r t = -\frac{\Delta n}{R^2} t \qquad (2)$$

where f is the focal length, t the thickness of the plate, $\Delta n$ the refractive index variation and R the half-diameter of the lens. The issue with such a refractive index profile is achieving a short focal length (i.e., a huge V value). Since V is proportional to the index variation (Equation (2)), a short focal length as the focal length of the studied camera requires a large $\Delta n$ value. For a 7 mm focal length like in our situation, this type of design needs a gradient index with a maximum variation of $\Delta n = 0.11$ for a thickness of 1 mm. This is higher than what has been achieved previously with a femtosecond laser [15]. One of the highest index variations reached to date in mid-IR glasses and with FLDW was about 0.05 in $As_2S_3$ (a chalcogenide glass) [32].

### 2.2. GRIN Fresnel Lenses

In order to reduce the index variation and preserve the small thickness of this component, the classical GRIN lens can be substituted with a Fresnel GRIN lens. In a classical Fresnel, the continuous surface of the lens is cut into a series of discontinuous rings [33,34]. Similarly, a GRIN Fresnel lens works on the same principle as the classical version: the continuous radial variation of the refractive index is cut into a series of discontinuous rings. The Fresnel versions of a spherical lens and a GRIN lens are shown in Figure 2.

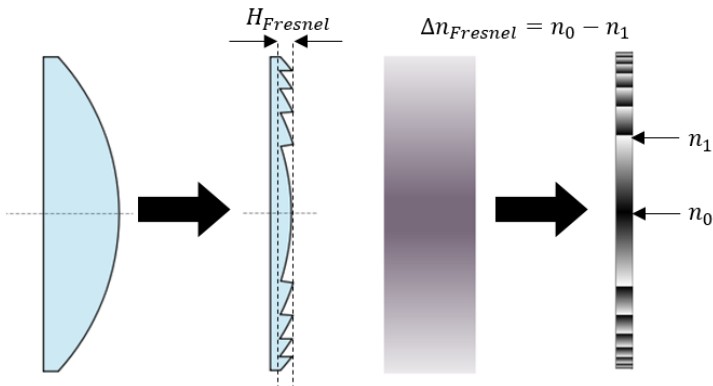

**Figure 2.** From a classical lens to a Fresnel lens and from a GRIN lens to a GRIN Fresnel lens.

A GRIN lens works similarly to a conventional lens, and the radial index variation corresponds to the radial variation of the sag surface for a conventional lens. The surface sag ($z_{Fresnel}$) of a lens is defined by its curvature and its conicity. The surface sag of a Fresnel lens is described in Equation (3) [35], and it corresponds to the sag surface of a conventional lens that is divided into a series of rings.

$$\begin{cases} z_{Fresnel}(r) = \dfrac{cr^2}{1 + \sqrt{1-(1+k)c^2r^2}} \bmod H_{Fresnel} \\ H_{Fresnel} = \dfrac{\lambda_0}{n_0 - 1} \end{cases} \tag{3}$$

where c is the curvature, k is the conic constant, mod is the modulo operation, $\lambda_0$ is the central wavelength and $H_{Fresnel}$ is the etching depth of a first-order Fresnel lens. In the case of a GRIN Fresnel lens, the surface sag is replaced by an index profile distribution ($n_{Fresnel}$) and so $H_{Fresnel}$ corresponds to the total index variation $\Delta n_{Fresnel}$:

$$\begin{cases} n_{Fresnel}(r) = \dfrac{n_r r^2}{1 + \sqrt{1-(1+k)n_r^2 t^2 r^2}} \bmod \Delta n_{Fresnel} \\ \Delta n_{Fresnel} = \dfrac{\lambda_0}{t} \end{cases} \tag{4}$$

where t is the thickness of the GRIN plate and $n_r$ is the coefficient of index variation (as in Equation (1)). The Zemax optical design software does not offer predefined Fresnel GRIN surfaces in its surface library. However, the user can specify a specific surface by using the "User Define Surface" function. Thus, we created a GRIN Fresnel surface for which the refractive index profile is defined by Equation (4). This surface was then implemented in the Zemax software and we defined the base index of the material ($75(GeS_2) - 15(In_2S_3) - 10(CsCl)$) at n = 2.08. The GRIN Fresnel lens was then optimized to meet the specifications of the camera with a spectral bandwidth of $[3.2; 3.4]$ μm.

We obtained a GRIN lens with a diameter of 3.8 mm that was composed of 38 rings, with the outer ring having a width of approximately 24.8 μm. This 100 μm thick GRIN Fresnel lens had a refractive index variation coefficient of $n_r \sim -0.67$ and a conic constant of $k \sim -1.9$.

Only one channel is modeled and represented in Figure 3 because each channel was the same, with a small adaptation of the total index variation as a function of the central

wavelength of the filter. Figure 3 shows a good conservation of the modulation transfer function (MTF) over the whole field, indicating a good contrast restitution. Although the contrasts of the monochromatic MTF ($\lambda = 3.3$ µm) were lower than the diffraction limit, the monochromatic MTF showed good conservation of the contrasts up to the Nyquist frequency (greater than 0.3). The image quality of the optical channel was thus considered good. However, the polychromatic MTF ($\lambda \in [3.2; 3.4]$ µm) dropped the contrasts at Nyquist frequencies below 0.1 due to the huge chromaticity of the first-order Fresnel lens. We noted a drop in the MTF at low frequencies due to the different defocused diffraction orders which created a parasitic background that lowered the quality of the contrast.

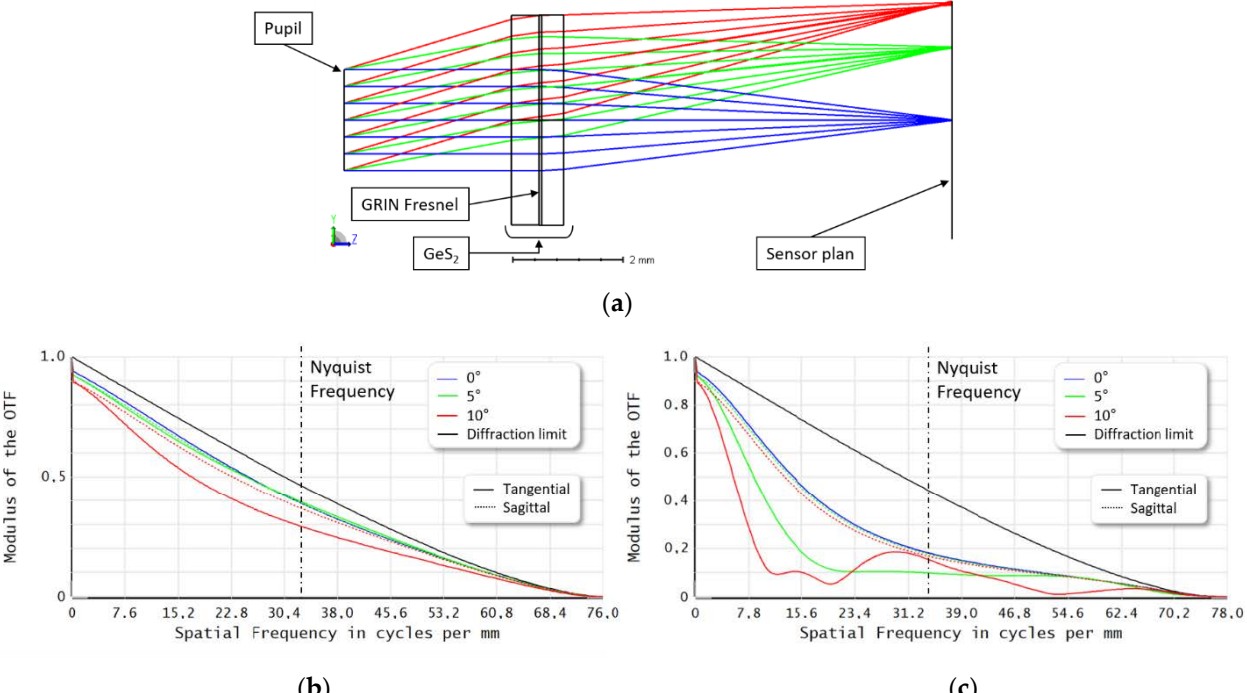

**Figure 3.** (**a**) Zemax 2D layout of a GRIN Fresnel lens with (from left to right) the pupil, the GeS$_2$ sample with a 100 µm thick GRIN Fresnel lens in the center of the sample and the image surface. Below, two Modulation Transfer Functions (MTF) given for three different half-fields of view ($0°$, $5°$ and $10°$). (**b**) The MTF is given for the central wavelength ($\lambda = 3.3$ µm). (**c**) The MTF is given for three different wavelengths of the spectral bandwidth: 3.2 µm, 3.3 µm and 3.4 µm.

To significantly increase the polychromatic MTF, the solution was to design a thirty-order Fresnel lens. However, manufacturing such a high-order Fresnel lens is difficult, so a first step was to work on manufacturing a first-order Fresnel lens and to test it under monochromatic illumination.

The fs-laser direct writing technique discretizes the profile into several index levels, so we needed to determine a compromise between the imaging quality of the Fresnel plate and the spatial resolution of the writing technique that discretized the variation of the index of refraction. For a first-order Fresnel lens as studied herein, we can define the diffraction efficiency as a function of the index versus wavelength variation (Equation (5)) [36].

$$\begin{cases} \eta^N_{m,\lambda} = \operatorname{sinc}^2[\alpha(\lambda) - m]\operatorname{sinc}^2\left[\frac{\lambda_0}{\lambda N}\right] \\ \alpha(\lambda) = p\frac{\lambda_0}{\lambda}\frac{n(r,\lambda)}{n(r,\lambda_0)} \end{cases} \tag{5}$$

where p is the Fresnel order, m is the considered diffraction order, $\lambda_0$ is the reference wavelength, N is the number of discretization levels and $n(r, \lambda)$ is the refractive index at the radial position r and the wavelength $\lambda$.

Figure 4 shows a good diffraction efficiency (up to 90% for the [3.2; 3.4] µm spectral bandwidth) for a first-order Fresnel lens discretized into eight index levels. Sixteen levels of discretization might be better, but would require rings thinner than 1 µm, which cannot be achieved with our laser direct writing technique. Indeed, with our setup, the laser waist (i.e., the writing spatial resolution) in $GeS_2$ is about 1.1 µm.

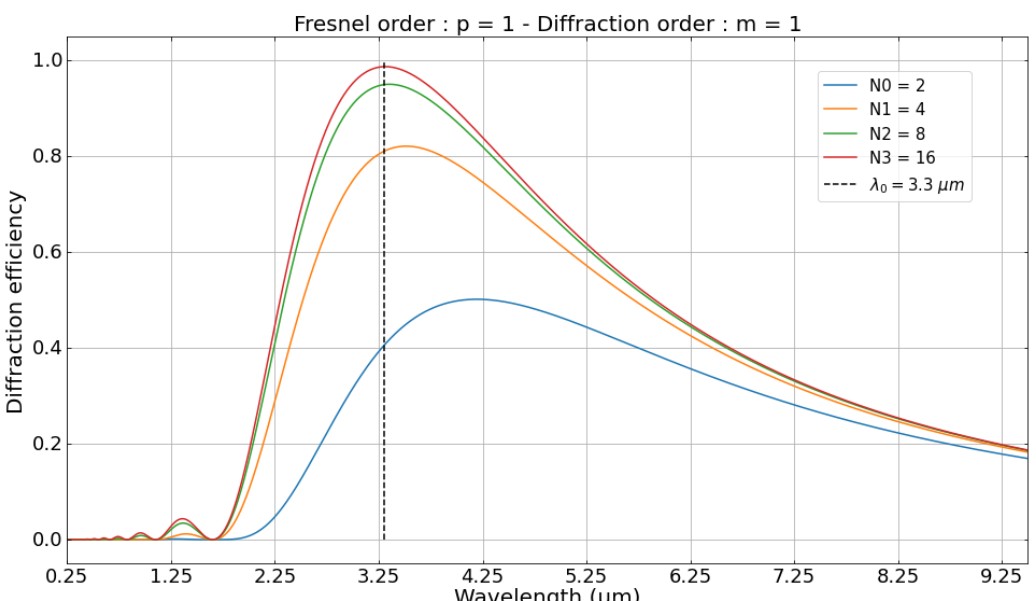

**Figure 4.** Graphic representation of the diffraction efficiency as a function of the wavelength for the designed first-order Fresnel lens at wavelength 3.3 µm. Only the diffraction efficiency of the first order of diffraction is shown. The different colors represent the different discretization levels. The dashed line represents the central wavelength of this design.

We modeled a GRIN Fresnel lens in our $GeS_2$-based material, which is presented in Figure 5. For 100 µm thick laser writing, the total gradient index was about $3.3 \times 10^{-2}$, which was achievable with our laser writing technique. With the numerical aperture specification of the imaging system, the last ring needed was about 24.8 µm thick, so we needed a lateral spatial resolution of about 3.1 µm to be able to write this last ring in such a way that we could discretize it into eight levels. Indeed, at this step, we needed to discretize this phase profile because our fs-laser direct writing software can only work at constant power, whereas ideally it would be preferable to have it continuously variable along the trajectory.

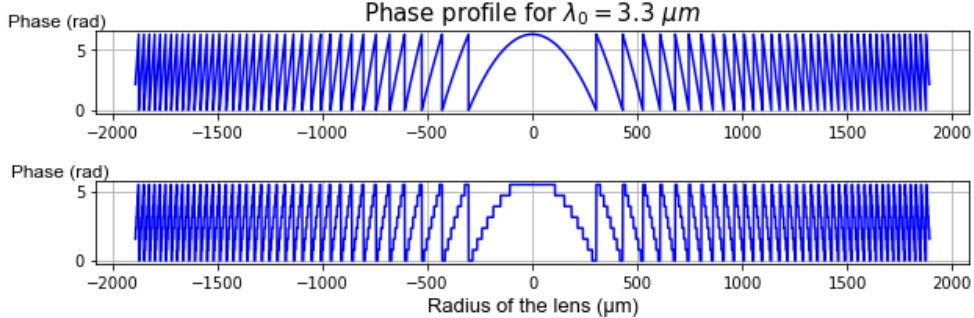

**Figure 5.** Refractive index profile of the GRIN Fresnel lens made in the chalcogenide glass $75(GeS_2) - 15(In_2S_3) - 10(CsCl)$: blazed on the top and discretized on the bottom.

### 3. Femtosecond Laser Direct Writing of Gradient Index Flat Lens

*3.1. Experimental Details*

*Glass:* In this work, we choose to use the $GeS_2$-based chalcogenide glass $75(GeS_2) - 15(In_2S_3) - 10(CsCl)$. At first, we noted that pure $GeS_2$ glass has a quite high glass transition temperature $T_g$ of 450 °C and is composed of repeating tetrahedral $GeS^{4/2}$ units. These units are predominately bonded at their corners to other $GeS^{4/2}$ tetrahedra, often labeled as $Ge^4$ (the superscript describes the number of bridging sulfurs), making up a fully connected three-dimensional structure. With its high glass transition temperature ($T_g \sim 370$ °C for our glass), we expect a quite long lifetime at room temperature for the photo-induced index changes [37]. Moreover, this glass presents a wide transmission band which can span across visible and LWIR windows. Indeed, this material has a transmission higher than 70% in the spectral band $[0.5; 10]$ μm for a 1 mm thick sample.

*Fs-laser Irradiation:* The glass samples were individually polished on both sides and placed on a three-axis motorized platform to be irradiated. The laser we used is a commercial Yb-doped fiber amplifier fs-laser (Satsuma, Amplitude Systèmes Ltd., Pessac, France). The key parameters were fixed as follows: a pulse duration of 800 fs, a central wavelength of 1030 nm and a repetition rate of 100 kHz. The laser polarization was linear and aligned along the *x* axis. The laser beam was focused inside the sample at 500 μm below its front face using an aspheric lens with a numerical aperture of 0.6, resulting in a spot diameter estimated at around 1.2 μm. The scanning speed of the sample, controlled by a motorized translation stage, and the laser energy, were varied. To write homogeneous disks or rings and to avoid any diffraction effects, the displacement of the sample followed a spiral trajectory with a step of 1 μm. The writing speed varied from 0.05 mm/s to 25 mm/s and the laser energy from 0.01 μJ to 0.75 μJ.

*Following the laser irradiation:* Each sample was placed on an Olympus BX51 polarized optical microscope. In addition to a typical optical microscope observation, index changes in writing kinetics were investigated using quantitative phase microscopy (QPM, Iatia, Victoria, Australia). This technique measures the phase retardation ($\Delta\varphi$, in rad) caused by the irradiated object with respect to the surrounding pristine glass. From these measurements, the refractive index modifications can be calculated through the following relationship: $\Delta n = \Delta\varphi\lambda/2\pi d$, where $\lambda$ is the wavelength of the illuminating microscope light (550 nm) and d is the thickness of the modified region. Additionally, room-temperature optical transmission measurements in the UV–Visible–NIR and the IR ranges were obtained, respectively, using Agilent Cary 5000 ($\Delta\lambda \in [0.2; 3.3]$ μm) and Bruker Equinox 55 spectrophotometers ($\Delta\lambda \in [2; 20]$ μm).

*3.2. Identification of Different Kinds of Permanent Modifications in $GeS_2$-Based Glass*

The FLDW technique locally modifies the physicochemical properties of a transparent material initiated by nonlinear absorption of laser pulse energy. Depending on the writing parameters, different kinds of modifications can appear in such optical glasses. These permanent changes are commonly classified in the literature into different types [38–40]: Type I, Type II and Type III. Type I corresponds to a smooth and homogeneous variation of the refractive index. This type of variation appears at a low energy threshold and can be simply seen through contrast using optical microscopy in natural light. This permanent and isotropic variation of the refractive index is very interesting for imaging applications because it allows precise control of the spatial refractive index distribution, associated with a fine tunability of the index modulation amplitude. Type II modifications are characterized by the appearance of an anisotropy in the index variation, the orientation of which can be controlled by laser polarization. This can be easily revealed using polarized optical microscopy and the full waveplate technique. Indeed, in some glasses (mostly oxide ones), specific conditions of pulse duration, repetition rate and pulse energy can create some birefringent self-assembled quasiperiodic nanostructures. Subsequently, beyond certain laser intensity, Type III appears, characterized by the presence of voids resulting from micro-explosions. Finally, another regime of modification appears, especially when

using a high-repetition-rate laser. It is often described inaccurately in the literature as a "heat accumulation regime", although it appears at low repetition rates as well when the deposited energy is quite high. This modification is not really a new type of modification because it is an isotropic variation of the refractive index like Type I. The difference is a significant increase in the size of the written area compared to the size of the laser spot due to heat diffusion. This regime is usually related to the pulse-to-pulse time interval (1/f, f being the repetition rate in Hz) being shorter than the heat diffusion time $\tau_{min}$ needed to evacuate the heat from each pulse. $\tau_{min}$ can be approximated using the following formula: $\tau_{min} \sim w_0^2/(4.D_{th})$, where $w_0$ is the beam waist radius, $D_{th}$ the thermal diffusion coefficient, expressed as $D_{th} = \kappa/(\rho \cdot C_p)$, where $\kappa$ is the thermal conductivity, $\rho$ the volumetric mass density and $C_p$ the specific heat capacity. In this paper, we call this regime the "spatial broadening regime" [25].

Thus, discrimination between what has been called Type I and spatial broadening relies on visual measurements based on a partly subjective criterion. Usually, the size of Type I modifications only slightly depends on the pulse energy and it remains compatible with the estimated spot size. This contrasts with a spatial broadening regime, where the heat-affected zone increases with the pulse energy, resulting in tracks much bigger than the laser spot size. In the following, we define the transition between Type I modifications and a spatial broadening regime as being an increase of the object by about half of its "initial value". Figure 6 shows the processing windows related to different types of modifications obtained as a function of the scanning speed and the laser pulse energy in $75(GeS_2) - 15(In_2S_3) - 10(CsCl)$ glass. Two examples of typical laser track cross-sections taken in optical transmission are also given for both Type I and the spatial broadening regime. The faster the writing, the higher the energy needed to write a permanent optical change into a glass substrate. We note that our material has a huge range of energies leading to type I changes even for high writing speeds up to 25 mm/s (a useful range of speeds for applications). This confirms that femtosecond laser direct writing is able to write GRIN optics and in a reasonable amount of time.

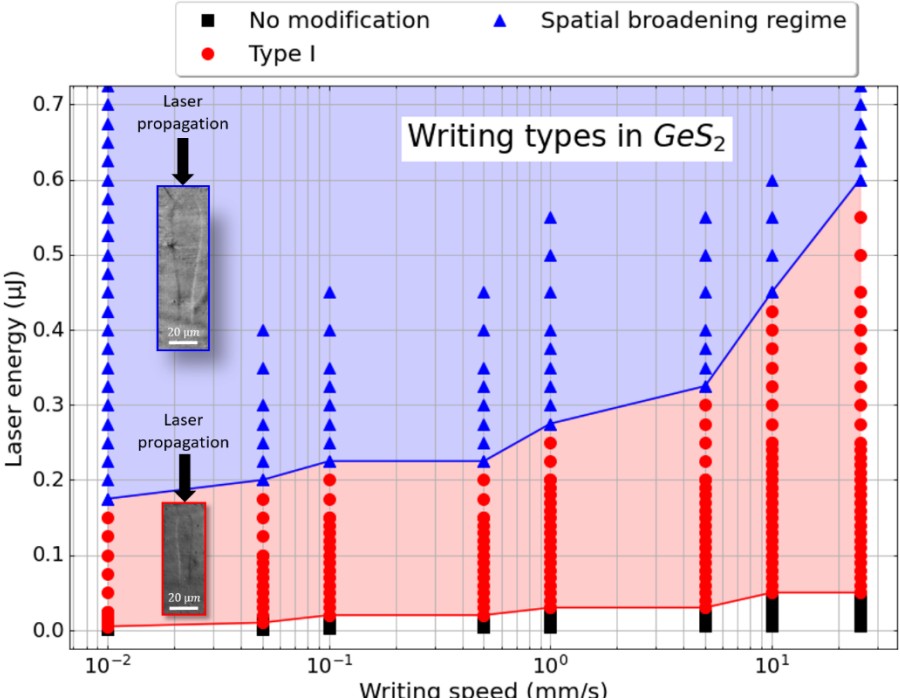

**Figure 6.** Processing windows in the $75(GeS_2) - 15(In_2S_3) - 10(CsCl)$ as a function of laser energy and writing speed. These values were found with a pulse duration of 800 fs and a repetition rate of 100 kHz. In the blue area: an optical microscope image of a laser track cross-section in the spatial broadening regime. In the red area: optical microscope image of a Type I cross-section.

Note that the spatial broadening regime could be a way to enable faster writing, as is the case for 3D optical waveguides (speeds up to 30 cm/s were demonstrated in a Gorilla glass [41]), or to write lines bigger than the laser beam waist, which allows the writing to be sped up for larger surfaces at the cost of a lower spatial resolution.

### 3.3. Phase Variation with Laser Energy and Writing Speed

GRIN Fresnel optics are more easily described by their phase variation rather than by their refractive index changes because these Fresnel optics are defined by a phase variation $\Delta\varphi = 2\pi$ rad for a first-order Fresnel lens. To write a GRIN Fresnel plate, we needed to know the precise phase variation implemented by the fs-laser as a function of the writing speed and the laser pulse energy. Therefore, several laser-written disks were made at varying energies and speeds within the type I regime. The phase variations of these disks were then measured using Quantitative Phase Microscopy (QPM) [42], and the results are reported in Figure 7 as a function of laser pulse energy. This technique consists of using the transport of intensity equation (TIE), which relies on intensity variation and phase variation. Using an optical microscope, a set of images was taken: one in-focus and two very slightly (a few microns) positively/negatively defocused [43]. Phase images given by QPM of the three laser-written simple disks are shown in Figure 7.

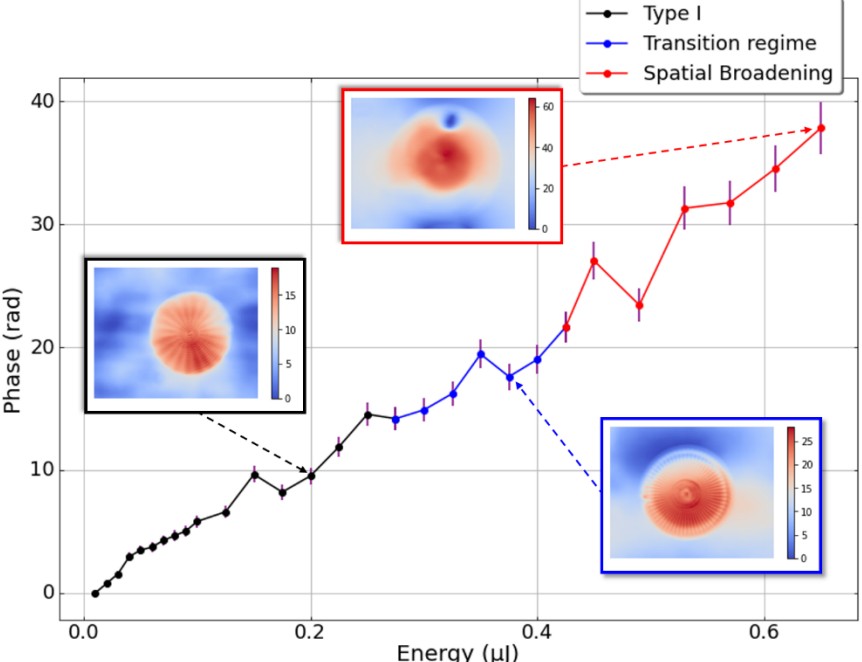

**Figure 7.** Phase variation implemented by fs-laser direct writing at a writing speed of 1 mm/s. Inset: quantitative phase map examples measured in $75(GeS_2) - 15(In_2S_3) - 10(CsCl)$ using QPM technique for the different regimes (Type I, transition regime and spatial broadening regime). *Parameters: 1030 nm, 800 fs, 100 kHz, 0.6 NA, 1 mm/s, linear polarization. QPM wavelength: 550 nm.*

We notice that a strong phase variation appeared at very low energy in our material, illustrating the technique's capability to write a significant phase variation even at 1 mm/s scanning speed. Higher writing speeds were investigated (up to 25 mm/s, as shown in Figure 6) with success, but a dedicated setup (higher acceleration stages or beam steering mirrors) would be more appropriate for accurately writing such complex 3D shapes at high speed. Therefore, we mostly used a 1 mm/s writing speed in our investigations.

The results presented in Figure 7 confirm a very high photosensitivity of $GeS_2$-based chalcogenide glasses to fs-laser writing. Indeed, phase changes up to 38 rad were implemented. A typical laser track length of around 100 µm corresponds to a refractive index change $\Delta n$ of about $5.5 \times 10^{-2}$ with respect to pristine glass. Preliminary annealing

tests revealed that a thermal treatment of 3 h at 300 °C (typically around $0.8T_g$ for our $75(GeS_2) - 15(In_2S_3) - 10(CsCl)$ sample) led to a 30% erasure only, which was mostly attributed to stress relaxation and related photo-elastic changes. In these conditions, we can expect a lifetime of at least 10 years at 50 °C with no significant decay (below 1% erasure) of the refractive profile, provided the optical component has been previously stabilized to erase the unstable part of the refractive index changes [37].

### 3.4. Optical Losses Pre- and Post-Irradiation

Another important feature when developing such a new manufacturing approach is the optical losses induced by the FLDW technique. Consequently, we measured the optical transmission ($\Delta\lambda \in [0.2; 20]$ µm) of our material before and after laser irradiation, using a 1.5 mm diameter disk as a test sample. Here, we chose the laser energy from the results given in Figure 7 to implement a $\pi$ rad phase change at 550 nm for the sake of simplicity. Indeed, $\pi$ rad is a basic phase shift value particularly used in phase mask optics (binary annular phase masks, binary Fresnel lenses, etc.).

Figure 8 represents the spectral transmission of the $GeS_2$ glass material pre- and post-irradiation. We noticed that the difference before and after irradiation was quite negligible (typically < 1% transmission change). As a result, the writing of a disk with a phase shift of $\pi$ rad did not affect the transmission. The results shown in Figure 8 allowed us to consider good conservation of the optical transmission even after imprinting some significant refractive index changes, even if, obviously, a higher phase variation will surely imply a more significant decrease of the optical transmission of the related optical components, mostly related to significant light scattering effects.

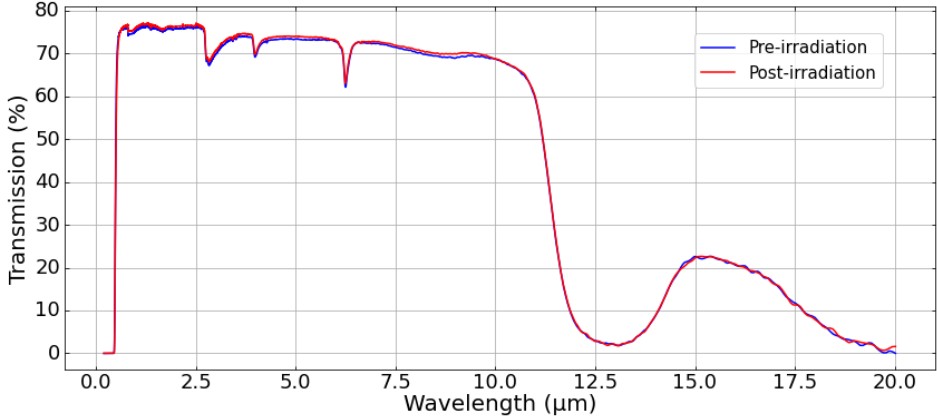

**Figure 8.** Spectral transmission in Vis–IR range before and after writing a 1.5 mm diameter homogenous disk with a fixed phase variation of $\pi$ rad in the $75(GeS_2) - 15(In_2S_3) - 10(CsCl)$ glass material. *Parameters: 1030 nm, 800 fs, 100 kHz, 0.6 NA, 1 mm/s, 0.05 µJ.*

## 4. Fabrication and Characterization of a Fresnel GRIN Plate

As described in the present section, an initial prototype of the Fresnel lens was written by femtosecond laser. We wanted to verify the ability of the FLDW technique to quickly write Fresnel lenses. The key laser parameters were fixed as follows: a pulse duration of 800 fs, a central wavelength of 1030 nm, a repetition rate of 100 kHz and a scanning speed of 1 mm/s. The laser beam was focused inside the sample at 500 µm below its front face using a microscope objective with a numerical aperture of 0.6.

There are two different ways to manufacture what we call fs-laser-written GRIN lenses. The first is described herein as the gradient thickness lens. It consists of creating a single value of refractive index variation $\Delta n$ and superimposing layers of uniform index changes to create a lens in the bulk of the material. Most previous papers on femtosecond laser graded index lens fabrication have dealt with this method [16]. The second is really a gradient index lens, meaning that the refractive index is varied spatially in the lens by changing the laser power during the writing process i.e., during the trajectory. Figure 9

describes the differences between these two manufacturing methods. It shows the gradient thickness lens written to index $n_1 = n_0 + \Delta n$ folded into the $n_0$ refractive index of the background material. In contrast, the gradient index lens showed a radial variation of the refractive index i.e., $n(r) = n_0 + \Delta n(r)$. These two lenses are optically equivalent in the basic phase profile but are manufactured differently.

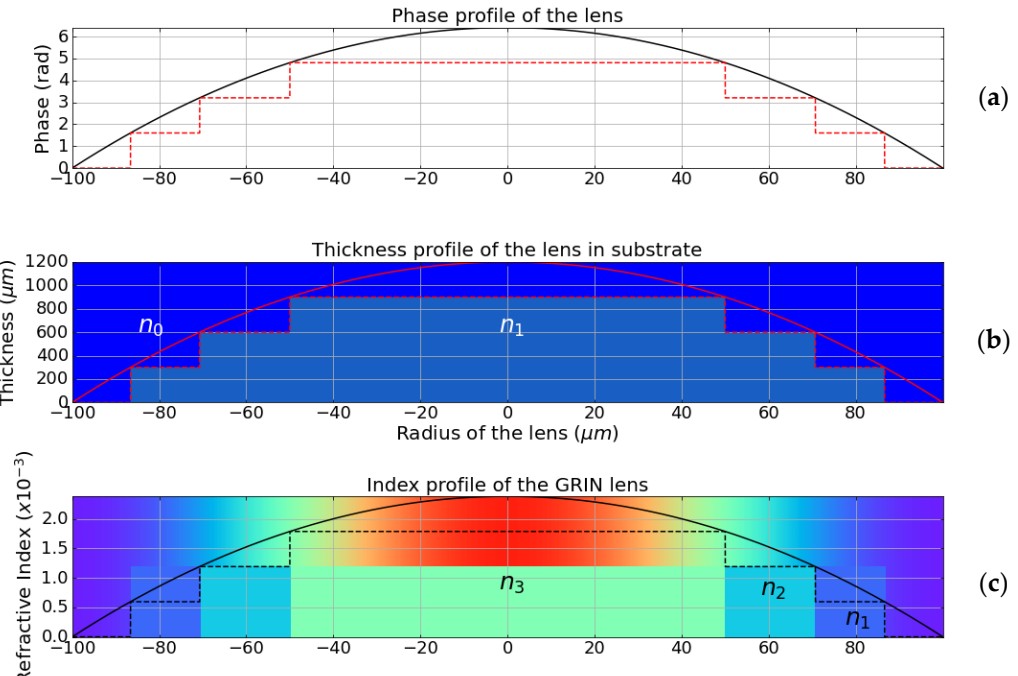

**Figure 9.** (**a**) The phase profile of a GRIN lens. (**b**) The corresponding thickness profile of a gradient thickness lens and its discretization. (**c**) The corresponding refractive index profile of a gradient index lens and its discretization. The color variation represents the radial refractive index variation. (**a**–**c**) The solid line represent the blazed profile and the dashed-line represent the discretized profile.

The thickness gradient lens is simpler to manufacture because only one value of laser energy is needed to obtain a specific variation of index $\Delta n$. The shape of the lens can then be created with constant energy. However, the gradient index lens allows manufacturing optics much more quickly since it can be achieved in a single layer. Indeed, once the curve (Figure 7) giving the variation of phase according to the laser energy is known, we only need to radially vary the energy to create the desired phase shift. Instead of a superposition of N discrete layers, which needs a lot of time, one layer with N energies is sufficient, or even a continuous (but controlled) change of the energy along the writing trajectory.

The gradient index lens method was used as it the fastest inscription process. We designed this Fresnel lens with the same characteristics as the GRIN Fresnel lens presented above (Figure 5), except that we chose a central wavelength of 550 nm due to the lack of available IR equipment and in order to simply measure the different focal lengths of the lens with a classical visible microscope (Olympus BX51) and some spectral filters. In addition, the diameter of the lens was reduced to only 500 µm (for a focal length of 2 mm) in order to write the Fresnel lens more quickly. The curve of phase variation as a function of laser energy (Figure 7) allowed us to write a first-order Fresnel lens in the GeS$_2$ glass, as presented in Figure 10.

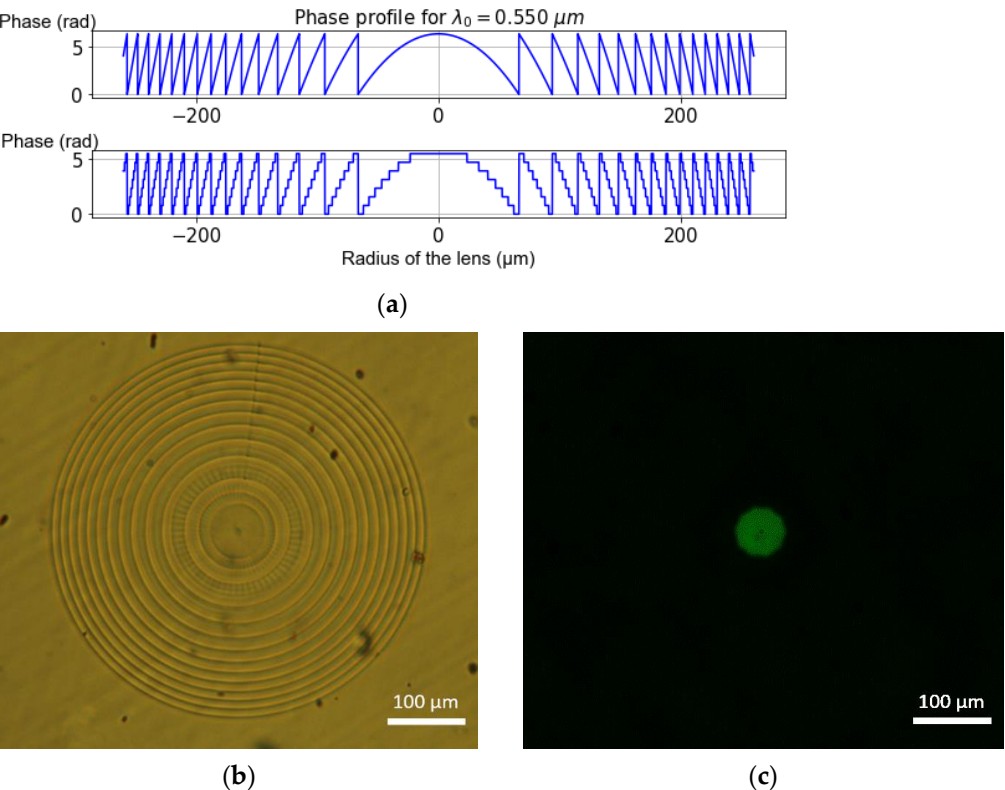

**Figure 10.** (**a**) The phase profile designed at a central wavelength of 550 nm. (**b**) Microscopic photograph of the graded index Fresnel lens written using the femtosecond laser direct writing technique (×4 objective). (**c**) Image of the field diaphragm of the microscope by the Fresnel lens at the first order (×10 objective, 550 nm filter).

These preliminary results were easily transposable from a Vis to an infrared system to build the final demonstration of our multispectral camera. Using the same optical material (our GeS$_2$-based chalcogenide) and an identical variation of index Δn, we achieved the same optical path difference (OPD). The phase difference then depended on the wavelength as follows:

$$\Delta\varphi(\lambda_{\text{IR}}) = \frac{2\pi}{\lambda_{\text{IR}}} \cdot \text{OPD} = \Delta\varphi(\lambda_0) \cdot \frac{\lambda_0}{\lambda_{\text{IR}}} \tag{6}$$

where e.g., $\lambda_{\text{IR}} = 3.3$ μm is the desired wavelength in the infrared range and $\lambda_0 = 0.55$ μm is the chosen wavelength in the visible range. We neglected the material dispersion here, assuming near conservation of the material dispersion when written by fs-laser (to be investigated in the future). Finally, we chose the diameter of this Fresnel lens to have a similar spatial resolution (about 3 μm) in the visible range to the Fresnel lens designed in the mid-infrared.

To characterize this Fresnel lens, we started by measuring the different focal points due to the different diffraction orders of this lens. For a first-order Fresnel lens with a gradient index, the focal length $f_m$ associated with the diffraction order m is [34]

$$f_m(\lambda) = \frac{r_1^2}{4m\lambda} \tag{7}$$

where $r_1$ the radius of the first Fresnel zone of the lens (in our case about 330 μm). The measurement of the different focal lengths of the plate was done with a microscope and spectral filter with a spectral band centered on 550 nm. At this wavelength, the focal length of the first order of diffraction (the most important for a first-order Fresnel lens) was $f_1(\lambda) = 7$ mm. Figure 10 shows an image of the field diaphragm of the microscope by the Fresnel lens positioned at about 7 mm from the lens.

We wanted to measure the variation of the focal length f as a function of the different orders of diffraction of the laser written lens and also the spectral dependence of the focal shift $\Delta f/f$. For these measurements, we used an Olympus BX51 visible microscope to which we added different visible filters with a spectral bandwidth of 10 nm (Thorlabs FKB-VIS-10). Figure 11 shows a comparison of the focal shift against the wavelength. We note that the image focal point was brighter for the first order of diffraction as we had designed our Fresnel lens to order 1. We measured three successive orders of diffraction with respective focal lengths of f, f/2, and f/3.

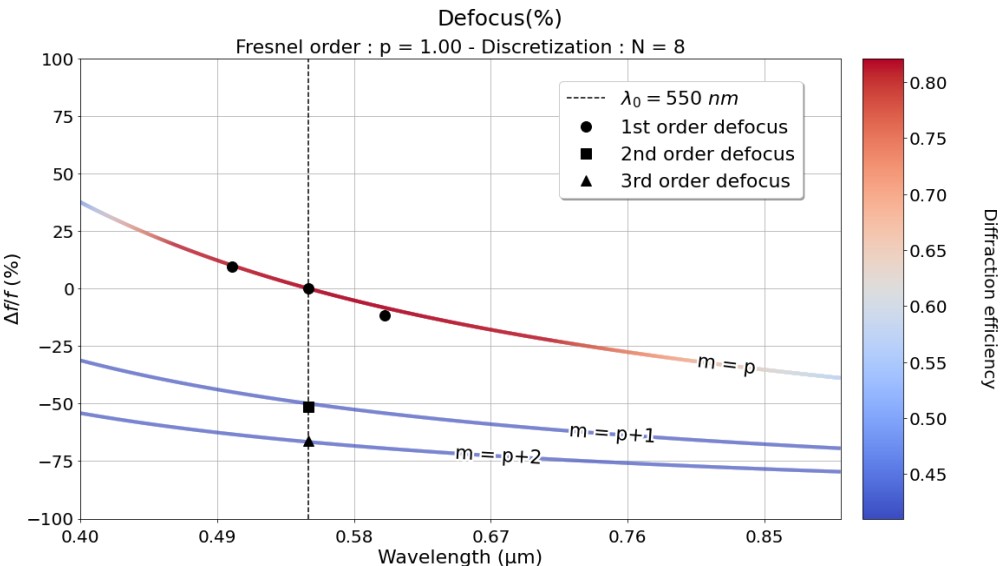

**Figure 11.** Focal length shift $\Delta f/f$ as a function of the wavelength for different orders of diffraction. Note that for the orders p + 1 and p + 2, the efficiency is not sufficient to see clearly the focal shift as a function of wavelength.

Based on Figure 11, the measurements of the defocus as a function of wavelength seemed to meet our expectations concerning the first order of diffraction. We note that the depth of field of the focal point was several tens of micrometers so it was difficult to accurately find the best focus. At this step, the losses associated with the diffraction efficiency of the Fresnel lens (typically between 40% and 70% at $\lambda_0$) and the losses of brightness to illuminate the object due to the addition of spectral filters did not allow us to measure the defocus of each diffraction order. Only the main focal lengths of these orders (f/2 and f/3) were measured and corresponded well to the expected value compared to order 1. A bench dedicated to the measurements of focal lengths and defocus of graded index lenses in the infrared is under preparation and will allow a more detailed analysis of the optical characteristics of these GRIN Fresnel lenses in the future. In addition, adding more layers (N) to the discretization will result in much higher diffraction efficiency in agreement with the theoretical considerations.

## 5. Conclusions

In this work, we designed and manufactured a graded index Fresnel lens with fs-laser direct writing and in a $GeS_2$ chalcogenide glass. We demonstrated the possibility of efficiently fabricating such GRIN Fresnel lenses with an initial working prototype designed to operate in the visible range (for simpler characterization purpose). A preliminary study of the variation of the focal length as a function of the wavelength allowed us to observe the expected chromaticity of this first-order Fresnel lens.

This work is a first step towards designing and manufacturing Fresnel lens arrays compatible with the cryogenic multispectral camera, where each channel works in a 200 nm broadband in the MWIR. We expect this technique to simplify the design of these cameras in a SWaP mitigation context. Higher-order Fresnel lenses may be needed to reduce the

strong chromaticity. The challenge will then be to find a compromise between the Fresnel lens order and the number of discretization levels in such a way that we can increase the diffraction efficiency while ensuring a reasonable fabrication time.

This study also confirmed the high photosensitivity of $GeS_2$-based chalcogenide glasses to fs-laser writing by implementing a refractive index variation higher than $5.5 \times 10^{-2}$. Thus, $75(GeS_2) - 15(In_2S_3) - 10(CsCl)$ glass seems very promising for femtosecond laser direct writing of complex optical components such as Fresnel lenses.

**Author Contributions:** Conceptualization, P.D., G.D. and M.L.; methodology, P.D., G.D. and M.L.; software, P.D.; validation, F.D.L.B., G.D., M.L. and L.C.; formal analysis, P.D.; investigation, P.D.; resources, G.D., M.L. and L.C.; data curation, P.D.; writing—original draft preparation, P.D.; writing—review and editing, P.D., F.D.L.B., G.D., M.L. and L.C.; visualization, G.D. and M.L.; supervision, F.D.L.B., G.D. and M.L.; project administration, M.L. and G.D.; funding acquisition, M.L. and G.D. All authors have read and agreed to the published version of the manuscript.

**Funding:** Agence Nationale de la Recherche (ANR-18-CE08-0004-01); FLAG-IR Project; Research supported by the Ministère des armées—Agence de l'innovation de défense (AID).

**Institutional Review Board Statement:** Not applicable.

**Informed Consent Statement:** Not applicable.

**Conflicts of Interest:** The authors declare no conflict of interest.

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
