# Peer review of "Femtosecond Laser Direct Writing of Gradient Index Fresnel Lens in GeS2-Based Chalcogenide Glass for Imaging Applications"

_applsci, doi:10.3390/app12094490_

Round 1

Reviewer 1 Report

This is an interesting paper that will be relevant to many people working on infrared optics.  There is a detailed explanation of the background and need to Fresnel GRIN lenses for IR imaging.  

The main criticism is that the final results are limited and poorly analysed compared with the design and modelling.  More experimental trials and data on written lenses is needed.  Particular some results on fabricated mid infrared lenses.  The title of the paper is misleading if it suggests that mid-infrared lenses have been written when they are in fact only for visible light.

The introduction does not give any values for SWaP.  What is too large and heavy and how much reduction is expected.

Specific chalcogenide glasses do not cover the visible and LWIR wavelength regions, their composition needs to be set for a specific band.

The authors should refer to some of the work by J Morris on laser written waveguides, for example.

3D optical waveguides in Ge22As20Se58 glass — a highly nonlinear material for the Mid-IR
James M. Morris;Giorgos Demetriou;Adam Lancaster;Ajoy K. Kar;Henry T. Bookey
2016 Conference on Lasers and Electro-Optics (CLEO)

In line 89 the mass of the filter and lens  needs to be stated and the improvement gained by the new device.  

Many IR cameras do not cool the lenses because they are transparent and so have low infrared emission.

There are a few minor English gramma improvements needed.

Line 100, please define vergence.

Line 145 "Although that the  contrasts"  Omitting  "that" expresses this better.

Line 227 "written geometrical are some disks of diameter 100 μm."  The gramma is unclear. 

line 164 It is not stated why the laser power is discretized rather than continuously variable.

Figure 6.  Please define how the writing types are defined and measured.

It is very surprising that the lens was written for a wavelength of 550nm since this is very close to the absorption edge of the glass.  A readily available camera and lens would have enable tests at 800-1000nm.

The summary of results on page 13 is brief and does not explain why the measured performance has high loss and poor diffraction efficiency compared with the models.  

The performance over wavelength is unclear if a filtering function is too be included in the lens.

The permanence of refractive index changes is a critical concern particularly against short wavelength exposure.  Please comment on this.

Reviewer 2 Report

General comments on the form of the document:

  • The manuscript does not present the required sections of the Applied Sciences journal format. Specifically, the document lacks the materials & methods, results, and discussion sections. Therefore, authors are recommended to adjust the text content to the sections of the magazine template.
  • It is recommended to check the order of the figures. For example, figure 4 and 5 should be placed after the paragraph in which they are mentioned.
  • It is appropriate to reduce the number of words in the abstract since it is higher than that allowed in the journal format (200 words).

Content Comments:

  • The document's content is similar to an article previously published by the authors (Pierre Delullier, Guillaume Druart, Florence de la Barrière, Laurent Calvez, and Matthieu Lancry "Femtosecond laser direct writing of a Fresnel zone plate in glasses for mid-infrared imaging applications", Proc. SPIE 11889, Optifab 2021, 1188912 (28 October 2021); https://doi.org/10.1117/12.2600695). In this sense, it is recommended to make a more significant differentiation between the content of the articles and to make the pertinent citations to the authors' previous works.

English language:

It is recommended to review in the text the use of:

  1. The articles. For example, in the sentence "Each channel has a focal length of 7 mm and a f-number of 4" is preferable to use the article an instead of a due to the phonetics of the word f-number.
  2. Prepositions. For example, in the segment "It consists in an array of 2×2 lenses with a focal length of 7 mm and an F-number of 4" it might be more appropriate to use the preposition of instead of in. Similar situations occur with the prepositions to, toward, and towards.
  3. Conjunctions. For example, in the sentence "Although that the contrasts of the monochromatic MTF" The conjunction "that" seems to be unnecessary.
  4. Use of the plural and the singular. For example, in the segment "Depending on the texturing parameters, different kind of modifications can appear in such optical glasses" the use of the word kinds, rather than kind is preferable.

Round 2

Reviewer 1 Report

As mentioned in the previous review some data to show the SWAP reduction would justify this work.  What is the weight reduction of the FLDW lens and how much does this reduce the cooling time?

How consistent is the change in refractive index due to annealing and are you able to compensate for this in the design processes.  Some people have observed a sensitivity to visible and UV radiation, the lenses may need protection from this.

Combining focusing and filtering are key aspects of this paper, but I can not find any evidence of filtering.